# Dietary Recommendations for Post-COVID-19 Syndrome

**DOI:** 10.3390/nu14061305

**Published:** 2022-03-20

**Authors:** Luigi Barrea, William B. Grant, Evelyn Frias-Toral, Claudia Vetrani, Ludovica Verde, Giulia de Alteriis, Annamaria Docimo, Silvia Savastano, Annamaria Colao, Giovanna Muscogiuri

**Affiliations:** 1Dipartimento di Scienze Umanistiche, Università Telematica Pegaso, 80143 Naples, Italy; luigi.barrea@unina.it; 2Italian Centre for the Care and Well-Being of Patients with Obesity (C.I.B.O), Department of Clinical Medicine and Surgery, University of Naples “Federico II”, 80131 Naples, Italy; metabolismounina@gmail.com (S.S.); operafederico2@gmail.com (A.C.); giovanna.muscogiuri@gmail.com (G.M.); 3Sunlight, Nutrition and Health Research Center, San Francisco, CA 94164-1603, USA; williamgrant08@comcast.net; 4School of Medicine, Universidad Católica Santiago de Guayaquil, Guayaquil 090615, Ecuador; evelyn.frias@cu.ucsg.edu.ec; 5Endocrinology Unit, Department of Clinical Medicine and Surgery, University of Naples “Federico II”, 80131 Naples, Italy; ludoverde96@gmail.com (L.V.); dealteriisgiulia@gmail.com (G.d.A.); annamariadocimo@gmail.com (A.D.); 6UNESCO Chair “Education for Health and Sustainable Development”, University of Naples “Federico II”, 80131 Naples, Italy

**Keywords:** COVID-19, post-COVID-19 syndrome, sarcopenia, nutrients, vitamin D, nutraceuticals, dietary recommendations

## Abstract

At the beginning of the coronavirus disease (COVID-19) pandemic, global efforts focused on containing the spread of the virus and avoiding contagion. Currently, it is evident that health professionals should deal with the overall health status of COVID-19 survivors. Indeed, novel findings have identified post-COVID-19 syndrome, which is characterized by malnutrition, loss of fat-free mass, and low-grade inflammation. In addition, the recovery might be complicated by persistent functional impairment (i.e., fatigue and muscle weakness, dysphagia, appetite loss, and taste/smell alterations) as well as psychological distress. Therefore, the appropriate evaluation of nutritional status (assessment of dietary intake, anthropometrics, and body composition) is one of the pillars in the management of these patients. On the other hand, personalized dietary recommendations represent the best strategy to ensure recovery. Therefore, this review aimed to collect available evidence on the role of nutrients and their supplementation in post-COVID-19 syndrome to provide a practical guideline to nutritionists to tailor dietary interventions for patients recovering from COVID-19 infections.

## 1. Introduction

Severe acute respiratory syndrome coronavirus 2 (SARS-CoV2) has spread rapidly to pandemic proportions since its first detection in Wuhan, China, in December 2019. As extensively reported, it causes a varied clinical spectrum, from asymptomatic infection to mild respiratory disease to severe multiorgan failure and death, referred to as coronavirus disease (COVID)-19 [1,2]. Beyond the effects on the lungs, there is now increasing knowledge of the interaction between cell metabolism and viral infection, which causes deleterious effects on inflammatory status, blood glucose control, and blood pressure [3]. On the other hand, there is mounting evidence that obesity and its complications (i.e., metabolic syndrome, insulin resistance, and type 2 diabetes) are significantly associated with the susceptibility to and severity of COVID-19 infection [4,5,6]. Consequently, multiple factors are involved in the prognosis and recovery from COVID-19 infections.

Acute illness and complications caused by COVID-19 have been extensively studied. However, by mid-2020, many studies reported that patients complained of symptoms persisting weeks after the acute illness. Currently, this condition is known as post-COVID-19 syndrome and is generally defined as “the persistence of signs and symptoms that develop during or following an infection consistent with COVID-19 which continue for more than 12 weeks and are not explained by an alternative diagnosis.” [7]. Large variation in the incidence and prevalence of post-COVID-19 syndrome exists. Nevertheless, it has been reported that post-COVID-19 syndrome is more frequent in hospitalized patients (85%) but also in individuals managed in the outpatient clinic or at home (10–35%) [8]. Therefore, given the rate of lasting COVID-19-related complications as reported in several studies [9,10,11], it is evident that post-COVID-19 syndrome will represent a burden for healthcare professionals and national health systems. 

In more detail, post-COVID-19 syndrome is characterized by a combination of symptoms, mainly fatigue and sleep disturbances [12]. Other common features are dyspnea, joint pains, anxiety, low mood, cognitive dysfunction, chest pain, thromboembolism, hair loss, and chronic kidney disease [12]. The pathophysiology of these symptoms could be related to direct viral damage, immunological/inflammatory sequelae, as well as an iatrogenic origin [13]. Over organ-specific effects, COVID-19 infection is known to cause severe catabolic muscle wasting [14]. Indeed, the significant systemic inflammation has negative effects on muscle protein synthesis, and there is increased nutritional demand, which is difficult to meet due to the loss of appetite, taste, and smell caused by the COVID-19 infection [15]. Therefore, skeletal muscle mass and function loss (sarcopenia), combined with poor intake due to frailty, low mood, and changes in the gut microbiome, have led to a high prevalence of malnutrition [15]. Malnutrition per se affects the recovery of all the other systems impacted by post-COVID-19 syndrome, so it is a key component that needs to be addressed. Consequently, nutritionists can play a crucial role both in the early onset of post-COVID-19 syndrome as well as in the follow-up of patients to improve the outcomes. 

Current scientific evidence provides extensive information about the physiopathological mechanisms behind post-COVID-19 syndrome, thus contributing to the identification of potential targets for nutritional interventions [16]. Nevertheless, the interpretation of the findings from several studies is challenging and might not offer a clear indication of the best solution in clinical practice. 

Therefore, this review summarized the available evidence on the role of nutrients and their supplementation on the main features related to post-COVID-19 syndrome. The aim of the present review was also to provide a practical guideline for the management of individuals with post-COVID-19 syndrome to be used in clinical practice. 

## 2. Methods

Literature searching for this review was conducted by searching the PubMed database for manuscripts (observational and clinical studies, systematic reviews, and meta-analyses) describing studies of human adults published in the English language, during the last 10 years. The terms “diet OR nutrition OR nutrients OR dietary recommendations OR nutraceuticals OR bioactive compounds”, “post-COVID-19 syndrome OR Coronavirus disease OR COVID-19 OR SARS-CoV2”, combined with the Boolean operator “AND”, were employed for the research. 

Complementary research was carried out when nonspecific evidence in patients with post-COVID-19 syndrome was available. In more detail, evidence on the role of nutritional management in patients with other diseases besides post-COVID-19 syndrome, with similar pathophysiological mechanisms and outcomes, was tentatively transferred to the treatment of clinical and physical complications complying with post-COVID-19 syndrome. 

This review included studies published in journals in the highest impact factor quartile in the “General Medicine” or “Endocrinology and Metabolism” or “Nutrition and Dietetics“ areas. We excluded all manuscripts not specifically related to each issue of interest.

## 3. Role of Nutrition in Patients with Post-COVID-19 Syndrome

Nutrition could play a key role in the management of post-COVID-19 syndrome. Indeed, multiple dietary compounds might have pleiotropic effects on different targets, thus contributing to alleviating symptoms and promoting both physical and psychological well-being, through independent as well as synergistic mechanisms, as reported below. 

### 3.1. Muscle Mass Restoration and Sarcopenia

Sarcopenia is a progressive and generalized condition that causes loss of muscle mass and function [17]. This disorder determines strength reduction, skeletal muscle failure, or insufficiency. The diagnosis is formulated with a stepwise approach. The first step consists in the measurement of the muscle strength by using the handgrip test. As a result of this test, sarcopenia can be excluded or suspected. Suspected sarcopenia needs to be confirmed by using other techniques that measure muscle mass, such as Dual-Energy X-ray Absorptiometry (DXA), Bioelectrical Impedance Analysis (BIA), Computed Tomography scan (CT), and Magnetic Resonance Imaging (MRI) [17].

Maintaining sufficient muscle mass and strength is important for healthy living. Acute sarcopenia is known to occur during COVID-19, especially in older patients, with direct implications for post-COVID-19 function and recovery [16]. Parenteral steroids used in patients with severe COVID-19 also contribute by increasing muscle protein breakdown [16]. Therefore, nutritional therapy to restore muscle mass is an important aspect in the management of post-COVID-19 syndrome. A recent systematic review and meta-analysis, evaluating nutritional interventions to improve muscle mass, muscle strength, and physical performance in older subjects, concluded that protein supplementation on top of resistance training can be used to increase muscle mass and muscle strength [18]. According to this review, the minimum daily protein requirement for healthy elderly subjects is 0.83 g of good-quality protein per kilogram body weight per day.

In addition, β-hydroxy-β-methylbutyrate or creatine supplementation was also effective, whilst the best evidence was available for the use of leucine, an essential amino acid [18]. This intervention—as supplements or in the form of a balanced diet—could apply also for muscle mass restoration in patients with post-COVID-19 syndrome, thereby favoring a faster physical recovery. 

### 3.2. Composition of the Gut Microbiota

The gut microbiota plays a unique metabolic function in the host. Indeed, reduced microbial diversity and gut dysbiosis have been recently implicated in host chronic diseases (i.e., inflammatory bowel disease, type 2 diabetes, cardiovascular disease, and colorectal cancer) [19]. Furthermore, the gut can communicate with the brain using neural, inflammatory, and hormonal signaling pathways, thus influencing psychological well-being [19].

Patients with COVID-19 have shown alterations in the composition of the gut microbiota, especially in the context of antibiotic use, and this can have both short- and long-term consequences for physical and psychological well-being, including recovery and the occurrence/severity of post-COVID-19 syndrome [20]. Studies have shown that protein, fats, digestible and non-digestible carbohydrates, probiotics, and polyphenols all induce shifts in the composition of the gut microbiota [21,22]. For example, a high-fat diet reduces Lactobacilli, which are known to be associated with a healthy metabolic state, while digestible and non-digestible carbohydrates enrich Bifidobacterium and suppress pathogenic Clostridia [21]. Furthermore, the use of both probiotics and polyphenols induces healthy changes in the gut microbiota [21,22]. Therefore, a balanced diet containing dietary compounds that support a healthy microbiota will help to promote physical and psychological well-being among patients with post-COVID-19 syndrome.

### 3.3. Post-COVID-19 Fatigue Syndrome

Recent studies showed that a significant proportion of COVID-19 patients suffer from prolonged post-COVID-19 fatigue syndrome, with symptoms resembling chronic fatigue syndrome (CFS) [23]. The pathophysiology is complex and involves autonomic dysfunction, endocrine disturbances, and reactive mood disorders (i.e., depression or anxiety), combined with genetic, environmental, and socio-economic predispositions [23]. At present, there is insufficient high-level evidence to directly support the use of nutritional supplements and modified diets to relieve symptoms in patients with post-COVID fatigue syndrome. However, there is evidence to support that the deficiency of some nutrients (i.e., vitamin C, vitamin B group, sodium, magnesium, zinc, folic acid, l-carnitine, l-tryptophan, essential fatty acids, and coenzyme Q10) seems to be important in the severity and progression of CFS symptoms by increasing oxidative stress [24]. Recently, several trials focusing on CFS patients have reported the benefit of antioxidants and lipids to reduce CFS symptoms. Indeed, the supplementation of glycophospholipid–antioxidant–vitamin demonstrated an improvement in the overall fatigue scores of moderate subjects measured using the Piper Fatigue Scale (PFS) [24]. Therefore, adequate nutritional supplements including essential fatty acids and antioxidants, or the same given in the form of a balanced healthy diet, could help in the control/alleviation of post-COVID-19 fatigue syndrome. 

### 3.4. Possible Role of Diet and Single Nutrients in Psychological Well-Being

In addition to the impact on physical health, post-COVID-19 syndrome also affects psychological well-being, including the development of anxiety, depression, post-traumatic stress disorder (PTSD), and cognitive impairment [25]. The association between nutrition and psychological well-being has gained increasing attention during recent years. Epidemiological studies have shown a reduced risk of depression associated with high fruit and vegetable intake [26]. Indeed, the supplementation of glycophospholipid–antioxidant–vitamin demonstrated an improvement in the overall fatigue scores of moderate subjects measured using the Piper Fatigue Scale (PFS) [26]. Furthermore, experimental exposure of healthy volunteers to diets with a high glycemic index has been shown to increase the occurrence of depressive symptoms [27]. In addition, studies have also shown that the Mediterranean diet can reduce markers of inflammation, while high intake of saturated and trans-fats and refined carbohydrates could result in cognitive decline and hippocampal dysfunction, leading to impaired psychological well-being [26,28]. Furthermore, it is worth mentioning that lipids constitute approximately 50–70% of the brain’s dry weight, and changes in the lipid environment of the brain result in functional alterations of the activities of receptors and other membrane proteins, with an impact on neurotransmission [29]. Notably, it has been demonstrated that diets rich in omega-3 fatty acids upregulated genes involved in maintaining synaptic function and plasticity in animals, and enhanced cognitive functioning in humans [29]. Additionally, omega-3 fatty acid deficiency is associated with an increased risk of developing various psychiatric disorders, and they are important for the maintenance of psychological well-being [29]. Therefore, it is evident that an overall healthy diet rich in fruits and vegetables, and bioactive compound constituents such as omega-3 fatty acids, with low intake of trans-fats and refined carbohydrates, can enhance psychological well-being, and, therefore, could play a role in recovery from post-COVID-19 syndrome. 

## 4. Role of the Nutritionist in the Management of Patients with Post-COVID-19 Syndrome

At the beginning of the COVID-19 pandemic, attention was focused on containing the spread of the virus and avoiding contagion. Now, it is evident that the focus should also be on the health status of COVID-19 survivors [30].

The role of the nutritionist within the multidisciplinary team is of outstanding importance. Indeed, it is well documented that the nutritional status of the patient largely determines the evolution of the COVID-19 infection [31], as well as in other conditions such as slowed wound healing, reduced organ function, cardiovascular diseases, and cancer, among others [32,33,34]. It is essential to note that the literature emphasizes the importance of formal nutritional evaluation for all patients with COVID-19 [35,36,37]. In fact, patients hospitalized for long periods tend to suffer from malnutrition and sarcopenia, mainly due to prolonged mechanical ventilation and immobility [38]. In France, the study by Vaillant et al. showed that among hospitalized patients who survived COVID-19, 67% presented malnutrition, which persisted in 41% after discharge [39].

Post-COVID-19 syndrome is more frequent if the patient’s nutrition is inadequate [36]. Apart from chronic fatigue, nutritional issues are related to the central sensitization that produces a hypersensitivity to stimuli, which has gastrointestinal implications, limiting food intake [40]. In addition, anorexia can be increased, and catabolism can be generated that further aggravates malnutrition and, therefore, the patient’s recovery [40]. Some patients with post-COVID-19 syndrome may require enteral or parenteral nutrition, while others will need qualified nutritional counseling to reverse malnutrition and/or treat pre-existing comorbidities [41]. In these patients, the nutritionist must perform a complete nutritional evaluation and, in this way, arrive at an accurate nutritional diagnosis.

One of the first actions that guides the nutritionists’ work is the application of nutritional screening; these tools allow them to identify those patients at risk of malnutrition [41]. One of the screenings used is the malnutrition screening tool. This instrument has the advantage of being quick to apply and determine a decrease in intake due to lack of appetite, and recent involuntary weight loss is promptly asked [42]. The Academy of Nutrition and Dietetics states that the evaluation by the nutritionist in these types of patients should include [41] the following.
(1)Nutritional history: nutrient intake (macro- and micronutrients), adequacy of energy and nutrient intake, past and current diet history, religious and cultural preferences, food intolerances and refusals, changes in appetite or habitual intake.(2)Anthropometric measurements: weight and height, to be able to perform the calculation of BMI, body compartment estimates (fat mass, fat-free mass). The Academy also recommends the analysis of biochemical data, in-depth physical examination, and inquiries about personal and family history, among other aspects. All of these details are important to establish a previous inflammatory state in these patients as it has been described that inflammation plays a determinant role in COVID-19 patients [43] and other diseases [44,45,46]. Apart from the BMI, it will also be essential to know the body composition. One of the techniques used is bioelectrical impedance analysis (BIA). It measures the bioimpedance to the flow of a low electric current at a fixed frequency, single or multi-frequency. The principle of this method is based on the fact that lean tissue is an excellent electrical conductor since it contains water and electrolytes, while fat is a poor conductor since it does not contain water [47]. This technique allows estimation of the percentage of fat mass and muscle mass, thus evaluating their variations over time.

As already mentioned, patients with post-COVID-19 syndrome present a high risk of developing sarcopenia. Various factors can explain this condition, including the previous medical and nutritional status and anorexia, low physical activity, cardiovascular complications, and the gut microbiota [48,49,50]. Studies carried out in people who suffered from previous coronaviruses show that, after infection, both physical function and physical form can be deteriorated for up to two years after having presented the disease [51]. Therefore, it is essential to evaluate this using various methods such as gait speed, handgrip, and questionnaires. One of the questionnaires used is the Strength, Assistance with walking, Rising from a chair, Climbing stairs, and Falls questionnaire (SARC-F): this tool proved to have a moderate–high specificity to accurately identify sarcopenia in older adults [52]. The patient must respond by rating these features (walking aid, strength, getting up from a chair, climbing stairs, and falls), then add the score and determine the presence or absence of sarcopenia [53]. Another widely used tool that indicates overall muscle strength, muscle mass, walking performance, and overall nutritional status is handgrip strength. A low handgrip strength indicates poor mobility and can be a good predictor of the clinical outcome of low muscle mass [54].

Studies in the general population and in patients who have had COVID-19 show that this pandemic also brought as a consequence poor quality of sleep and insomnia [55,56]. Sleep disorders can appear as a result of many other conditions, such as obesity [57], hypovitaminosis D [58], obstructive sleep apnea [59], hormonal and emotional disturbances, among others. Therefore, it is also crucial to assess sleep in patients with post-COVID-19 syndrome. To assess this, one of the widely used tools is the Pittsburgh sleep quality index [60]. This tool evaluates sleep quality and disturbances over a period of time (1 month). Different components of sleep are estimated, which then yield a score that can be distinguished between those who sleep well and those who do not sleep [61].

The literature is overwhelming on the importance of the nutritional approach among COVID-19 survivors. For this, the role of the nutritionist is fundamental; he/she must collect enough information through anamnesis, diagnostic tests, and screening to arrive at a nutritional diagnosis that allows actions to be taken to improve the state of health and quality of life of the person (Table 1).

## 5. Dietary Recommendations for Patients with Post-COVID-19 Syndrome

As reported in the previous sections, post-COVID-19 syndrome is characterized by malnutrition, loss of fat-free mass, and low-grade inflammation. In addition, the recovery might be complicated by persistent symptoms such as functional impairment (i.e., fatigue and muscle weakness), dysphagia (particularly in patients who were intubated during hospitalization), appetite loss, and taste/smell alterations (ageusia/dysgeusia and anosmia) [62]. Therefore, the goal of nutritional therapy in post-COVID-19 syndrome should focus on the correction of nutritional deficiencies to support an adequate recovery in terms of physical and functional conditions, as well as mental health. 

### 5.1. Energy Intake 

The energy requirements for patients with post-COVID-19 syndrome depend on their actual nutritional status. Most individuals experienced unintentional weight reduction during COVID-19 infection, due to increased inflammation, appetite loss linked to taste/smell alterations, and swallowing disorders. In addition, patients might present early satiety and fullness after eating and drinking [63,64,65]. Therefore, it is important to correct the imbalance between energy expenditure and energy intake. Over the estimation of individual energy requirements (according to age, gender, and weight), patients can be advised regarding practical strategies to increase their food intake, such as consuming smaller and more frequent meals (six meals/day, snacking every 3 hours), drinking away from meals to avoid early satiety, and limiting foods or beverages labeled “light”, “low fat”, or “low calorie”. Ready-to-drink, low-volume oral nutritional supplements might be considered to increase energy intake [66]. Nevertheless, some patients were overweight/obese before COVID-19 infection, whereas some individuals gained weight during confinement for variations in eating habits, stress, mental burden, and physical activity [67,68,69]. Evidence showed that patients with overweight/obesity present a greater risk of worse outcomes after acute infections [70], and are more prone to develop viral infections [71]. On the other hand, obesity is characterized by a pro-inflammatory state with increased outflow of inflammatory cytokines (i.e., IL-6 and tumor necrosis factor-alpha) [71]. Therefore, weight loss is advocated in patients with post-COVID-19 syndrome to prevent future viral infections and reduce obesity-linked sub-clinical inflammation. 

### 5.2. Macronutrients

The protein requirement should be higher in patients with post-COVID-19 syndrome to improve sarcopenia and avoid further wasting of muscle mass [72]. Patients should be recommended to include high-quality proteins, both from plant and animal sources, and to consume 15-30 g protein/meal, depending on body weight, to ensure their intake of all the essential amino acids, which might exert an anti-inflammatory effect [73,74]. Moreover, some studies suggested that consuming protein during the day might prevent autophagy [75]. Therefore, it could be useful to include a protein source at each meal and snack. In addition, certain amino acids, i.e., arginine and glutamine, might be supplemented by virtue of their known role in the modulation of the immune response [74]. 

As for fat intake, a daily intake of 1.5–3 g/day of omega-3 fatty acids (eicosapentaenoic acid and docosahexaenoic acid) should be advised to improve inflammation. Interestingly, it has been shown that omega-3 fatty acids might inhibit the viral replication of enveloped viruses—such as COVID-19—possibly reducing the risk of new infections [76]. Moreover, the consumption of extra-virgin olive oil should be increased to provide adequate intake of monounsaturated fatty acids, tocopherols, and polyphenols, which have demonstrated anti-inflammatory and antioxidant properties [77]. 

Finally, total carbohydrate intake is not a major concern in patients with post-COVID-19 syndrome. However, the consumption of carbohydrate sources with a low glycemic index is highly recommended. Indeed, the intake of food with a high glycemic index has been associated with increased inflammation and oxidative stress [78,79]. Furthermore, the intake of viscous and fermentable fibers (i.e., β-glucan and arabinoxylans from wholegrain, pectins from fruit, vegetables, and legumes) should be increased by virtue of its prebiotic effect towards butyrate-producing bacteria, which has been associated with reduced inflammation in the host [80,81]. 

### 5.3. Micronutrients

The role of nutrition, particularly trace elements and vitamins in modulating immunity, has received much interest during the pandemic. Indeed, a pilot study assessed micronutrient status in hospitalized patients for COVID-19 showing micronutrient deficiency, particularly in vitamin D (76%) and selenium (42%) [82]. 

The role of vitamin D in reducing infections is carried out through several mechanisms. These include the induction of cathelicidins and defensins, diminishing virus survival and replication, and keeping undamaged the epithelial layers [83]. Those specifically related to COVID-19 infection include the reduction of pro-inflammatory cytokine concentrations and augmenting ACE2 levels [84]. It should be noted that the effects of vitamin D supplementation on indices of inflammation and oxidative stress have also been studied in other diseases, such as diabetic patients [85,86] and hypertensive patients [87], as well as its effect on calcium metabolism and broad non-calcemic gene expression [88]. Another disease that has been related to low concentrations of 25-hydroxyvitamin D (25(OH)D) is breast cancer, seeing that higher concentrations of this vitamin are associated with a lower risk of breast cancer [89].

Notably, vitamin D deficiency has been reported to be common in patients with cardiovascular disease (CVD) [90,91]. The Mendelian randomization analyses performed in four population-based cohort studies found an inverse association between the results of coronary heart disease, stroke, and all-cause mortality with a low serum concentration of this vitamin [92]. Acharya et al. observed that in patients with vitamin D deficiency and no history of myocardial infarction, treatment with a certain level of 25(OH)D was associated with a significantly lower risk of mortality from all causes [93].

It is interesting to mention the meta-analysis carried out by Schöttker et al., where the association between serum concentrations of 25(OH)D and mortality was investigated in different cohort studies with individuals with differences by age, sex, and country. The researchers concluded that although vitamin D levels varied according to country, sex, and the year’s season, there was a consistent association between the level of 25(OH)D and all-cause and cause-specific mortality [94]. Therefore, it is evident the major impact of vitamin D levels in several health conditions that COVID-19 patients may have; thus, their prognosis will be poorer than those who do not have this deficiency.

Seal et al., in their cohort study published in 2022, observed that in a total of 4599 SARS-CoV-2 positive patients, in an inverse dose–response relationship, the blood concentrations of 25(OH)D were associated independently with hospitalization and mortality related to COVID-19 [95]. Similar results were obtained in other publications [96,97], which separately concluded that these also occurred in patients with comorbidities [98].

Therefore, the ESPEN guidelines [66] and others [99] recommend that patients should consume 100% of the recommended daily intake (RDA). Higher intakes would be required [100], and a multivitamin and mineral supplement should be advised at least once daily for patients with micronutrient deficiency. The international nutritional recommendation suggests the importance of vitamin D intake (400 IU), particularly in patients with lower exposure to sunlight (i.e., long-term confinement or hospitalization) [82,101].

### 5.4. Nutraceuticals 

Over the abovementioned nutrients, other bioactive compounds might have a pivotal role in reducing inflammation (immunosuppressants) or improving the immune response (immunostimulators). 

Among immunosuppressants, polyphenols (quercetin, resveratrol, catechins), N-acetyl-cysteine (NAC), and palmitoylethanolamide (PEA) have demonstrated their antiviral activities, mainly involving the inhibition of inflammatory pathways (i.e., NLRP3 inflammasome-mediated IL-beta production and pro-inflammatory cytokine secretion) as well as viral replication (through the inhibition of the main viral proteases) [102]. In addition, inositol supplementation may reduce the cytokine storm, which characterizes COVID-19 infection [100], possibly playing a key role also in the recovery process. On the other hand, increased inflammation is a consequence of impaired oxidative status. Over vitamins and minerals with antioxidant properties, glutathione supplementation improves oxidative damage in several tissues [103]. Therefore, the association of inositol and glutathione can represent a useful strategy to improve inflammation and oxidative status in patients with post-COVID-19 syndrome. 

As for immunostimulators, milk proteins and peptides (bovine lactoferrin, lactoperoxidase, serum albumin, β-lactoglobulin, and α-lactalbumin) have been used as effective immune boosters [102], although the mechanisms underlying this beneficial effect are not completely clear. Furthermore, probiotics (i.e., *Lactobacillus* and *Bifidobacteria*) might improve the immune response, favoring the competition with pathogens for colonization in the gut and maintaining intestinal barrier integrity, thus reducing permeability to a pathogen and its microbial metabolites [104]. 

Finally, some nutritional compounds have been proposed as immunomodulators for the treatment of COVID-19 or to attenuate its symptoms. Glychophosphopeptical (AM3), a glucan glycophosphopeptid, can modulate both innate and adaptive immunity [105]. *Polypodium leucotomos* extract is known for its pleiotropic effect on different pathways related to the immune response [106]. Glutamine is a conditionally essential amino acid that plays a critical role in the modulation of the “cytokine storm” during COVID-19 infection [107].

### 5.5. Mediterranean Diet

Mounting evidence demonstrates that dietary intake (including nutrients and non-nutritive bioactive compounds) could modulate inflammation and the immune system. Therefore, the combination of different foods with these properties in a whole dietary pattern can be used as a useful nutritional approach for patients with post-COVID-19 syndrome.

The Mediterranean diet is characterized by many bioactive compounds with anti- inflammatory and antioxidant activities (monounsaturated and omega-3 fatty acids, and vitamins, minerals, and phytochemicals, respectively) [50]. Indeed, several studies confirmed the anti-inflammatory and immunomodulatory effects of a Mediterranean diet on several diseases associated with chronic low-grade inflammation [108]. Interestingly, observational studies highlighted an association between adherence to the Mediterranean diet and better outcomes in patients with COVID-19 (mortality, recovery rate) as well as risk of COVID-19 infection in different populations [76,109,110,111,112] Therefore, it is recommended to consume more plant-based foods (fruit, vegetables, wholegrain, and legumes), high-quality animal proteins (fish, lean meat, poultry, eggs, and low-fat cheese), and extra-virgin olive oil as the principal source of fat [113]. 

Lastly, adequate hydration (30 mL/kg actual body weight) is important for the complete recovery of patients with post-COVID-19 syndrome [114]. Therefore, these patients should increase their daily fluid intake (2.5–3 L/day) by consuming water, milk, fruit juice, broth, sports drinks, coffee, and tea.

## 6. Conclusions 

In conclusion, patients with post-COVID-19 syndrome need a personalized evaluation of nutritional status to detect potential nutrient and non-nutrient deficiencies and to improve physical and mental complications and the overall health status. Patients should be advised to include several foods that naturally contain bioactive compounds with anti- inflammatory and immuno-stimulating activities (Figure 1).

The Mediterranean diet might be a useful strategy to achieve this purpose. Supplements and nutraceuticals should be advised in malnourished and deficient patients, and in those not adherent to nutritional recommendations for lasting physical complications linked to COVID-19 infections (Figure 1). 

Although the evidence on the nutritional management of patients with post-COVID-19 syndrome is still meager, all the proposed recommendations reported in this review might effectively influence the main pathophysiological mechanisms underlying post-COVID-19 syndrome (Figure 2). Nevertheless, it is important to underline that some information reported in the present review was obtained in studies addressing the treatment of diseases with similar outcomes and not specifically post-COVID-19 syndrome. Therefore, further studies involving patients with post-COVID-19 syndrome are required to provide the best clinical approach to face this novel disease. 

## Figures and Tables

**Figure 1 nutrients-14-01305-f001:**
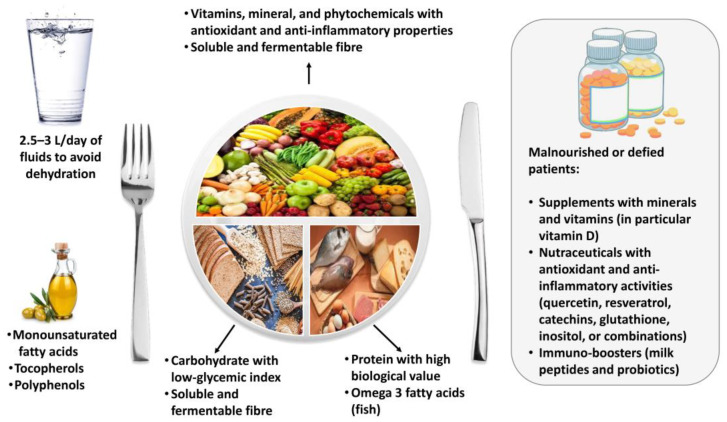
Dietary recommendations for patients with post-COVID-19 syndrome.

**Figure 2 nutrients-14-01305-f002:**
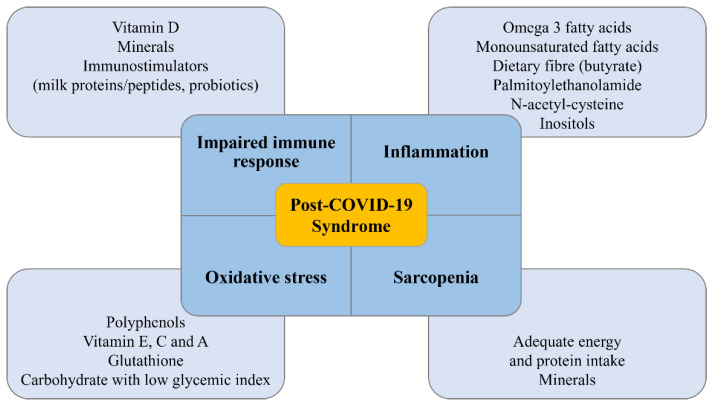
Main targets of recommended dietary compounds in patients with post-COVID-19 syndrome.

**Table 1 nutrients-14-01305-t001:** Summary table of the tools/procedures for nutritional assessment in patients with post-COVID-19 syndrome.

Target	Tool/Procedure
Risk of malnutrition	Malnutrition screening tool (dietary intake, appetite, and involuntary weight loss)
Dietary habits	Nutritional history (adequacy of actual energy and nutrient intake, religious and cultural preferences, food intolerances and refusals, past diet history, changes in habitual intake)
Anthropometry and body composition	-Body size (weight, height, and BMI) -Bioelectrical impedance analysis (fat mass, fat-free mass, and muscle mass)
Sarcopenia and functional impairment (fatigue and muscle weakness)	-Gait speed -Handgrip -Specific questionnaires (i.e., SARC-F)
Physical impairment	Anamnestic data (dysphagia, taste/smell alterations) Biochemical parameters (in particular, inflammatory status)
Sleep disorders	Pittsburgh sleep quality index

SARC-F: Strength, Assistance with walking, Rising from a chair, Climbing stairs, and Falls questionnaire.

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
