# Peer review of "Dietary Recommendations for Post-COVID-19 Syndrome"

_nutrients, 2022, doi:10.3390/nu14061305_

Round 1
Reviewer 1 Report
This narrative review presents very relevant recommendations for the nutritional recovery of patients with the post-covid19 syndrome (PC19S). The authors relied on several physiological derangements (e.g. sarcopenia, elderly frailty, delayed cytokine response, fatigue syndrome, gut dysbiosis) to objectively (sometimes supported on clinical findings) and subjectively (authors´ comments) suggest the need for specific (personalized) dietary/supplement interventions for people with PC19S. Based on the supporting evidence analyzed by the authors, it is my opinion that the proposed dietary recommendations and nutritional assessment strategies are general rather than PC19S-personalized, given that nutritional interventions for this syndrome are still scarce, if not null. Some modifications are suggested to improve the scientific soundness and uniqueness of the manuscript:
- Title. Suggestion: Dietary recommendations for post-CoVID-19 syndrome
- Introduction. Most, if not all, pathophysiologic deviations discussed in this section, belong to CoVID-19 inpatients or from those who died from this cause. The "pathophysiological signature" that characterizes PC19S in outpatients (discharged patients) should be perfectly stated and the public health burden of PC19S should be discussed in brief (see DOI: 1101/2021.01.15.21249885, 10.1371/journal.pone.0249644, 10.1371/journal.pone.0254523), highlighting the benefit of effective nutritional interventions.
- Body of text. A) A brief description of what is PC19S as a sequel of acute CoVID-19 should initiate the narrative review. Recognizing the lack of a general clinical consensus for PC19S, several publications may help to lay out an "inventory" of symptoms and complications that will allow for more specific nutritional recommendations for these outpatients (e.g. DOI: 10.1007/s12016-021-08848-3, 10.1016/j.lanepe.2021.100122, 10.1016/j.arcmed.2021.03.010, 10.1016/j.eclinm.2021.100899, 1016/S2213-2600(21)00125-9), B) Provide, when possible, specific information as to the personalized nutritional management of COVID-19 inpatients and outpatients (see DOI: 10.1186/s13027-021-00401-3) and, C) Reduce as much as possible unnecessary references, particularly multiple citations (e.g. line 62, ref. 10; line 88, ref. 3-5) or those related to COVID-19 inpatients but not outpatients.
- Tables. It is recommended to compile in Tables any evidence of the effectiveness of nutritional interventions in patients with PC19S (Table 1) and comprehensive nutritional assessment tools/procedures (Table 2) especially targeting all nutritional deviations commented.
- Figures. The recommendations in Figure 1 are very general and not specific. It is recommended to complement this figure with another one where the main physiological disorders associated with PC19S are reflected.
- References. They are too many, though very recent. Performing the suggested modifications to the discussion section may help to reduce the number of unnecessary references.
Reviewer 2 Report
The manuscript was prepared very well. However, there are some concerns, in part important, so the review articles need revision, see below.
General comments
I have carefully read the manuscript but there are still not enough studies or high degree of evidence that certify the possibility of treatment of long COVID with the nutrients that have been raised and the content is mere speculation. Therefore, I consider that it should be a hypothesis manuscript and not a review manuscript and should be completely rethought.
Introduction
In the introduction there are too many references to COVID-19. This knowledge is already well described after 2 years of pandemic. An overview of the long COVID and why nutritional intervention is necessary should be included.
Is there any pharmacological treatment? why implement nutrition or supplementation?
Materials and Methods
Could you include a brief search methodology for the studies used?
Results
One of the main problems of long COVID is the deregulation of the immune system, and no immunomodulator is proposed as a potential nutrient.
Although it proposes potentially effective nutrients for long COVID, it does not explain the potential restoration mechanism of the patient with long COVID. include it
What specifically does this manuscript contribute?
Include a limitations section.
Round 2
Reviewer 1 Report
Thank you for having accepted most of my comments, the article improved substantially
Author Response
Thank you again for your valuable suggestions.
Reviewer 2 Report
The authors have made substantial changes that have improved the manuscript. I congratulate the authors for this. The manuscript requires minimal changes and requires more extensive language editing.
Results
One of the main problems of long COVID is the deregulation of the immune system, and no immunomodulator is proposed as a potential nutrient.
Although there is no certainty about the use of immunomodulators, neither is there any certainty about the other nutrients you describe.
Some have been proposed for the treatment of COVID-19 or to attenuate its symptoms. You should extend this comment to Glycophosphopeptical (AM3), Polypodium leucotomos, Glutamine and Immuno Ft.
